Congruence between morphology-based species and Barcode Index Numbers (BINs) in Neotropical Eumaeini (Lycaenidae)

Prieto Carlos 1 2 cprieto50@gmail.com
http://orcid.org/0000-0001-9330-4633 Faynel Christophe 3
http://orcid.org/0000-0003-4137-786X Robbins Robert 4
http://orcid.org/0000-0002-0358-9928 Hausmann Axel 5
1 Departamento de Biología, Universidad del Atlántico , Barranquilla , Colombia
2 Corporación Universitaria Autónoma del Cauca , Popayán , Colombia
3 16 rue des Aspres, F-34160 Montaud , France
4 Department of Entomology, Smithsonian Institution , Washington , USA
5 SNSB-Bavarian State Collection of Zoology , Munich , Germany
Fernández Rosa M.
Electronic publication date: 2021 Aug 5
Publication date: 2021
Volume: 9
Electronic Location ID: e11843
Received 2021 Feb 9; Accepted 2021 Jul 1
Copyright: © 2021 Prieto et al.
Copyright year: 2021
Copyright holder: Prieto et al.
License: This is an open access article distributed under the terms of the Creative Commons Attribution License, which permits unrestricted use, distribution, reproduction and adaptation in any medium and for any purpose provided that it is properly attributed. For attribution, the original author(s), title, publication source (PeerJ) and either DOI or URL of the article must be cited.
License URL: https://creativecommons.org/licenses/by/4.0/

Keywords: Barcodes, Genetic library, Lepidoptera, Theclinae, Butterflies

Funding: Georg Forster Research Fellowship Program of the Alexander von Humboldt Foundation (Bonn) Research Group Linkage Programme Evolution of the high Andean insect fauna project, the Federal Ministry for Education and Research (Germany) Corporación Universitaria Autónoma del Cauca (Popayán, Colombia) Vice-Rectorate for Research of the Universidad del Atlántico, (Barranquilla, Colombia) This research was supported with funds from the Georg Forster Research Fellowship Program of the Alexander von Humboldt Foundation (Bonn), the Research Group Linkage Programme: Evolution of the high Andean insect fauna project, the Federal Ministry for Education and Research (Germany), the Corporación Universitaria Autónoma del Cauca (Popayán, Colombia) and the Vice-Rectorate for Research of the Universidad del Atlántico, (Barranquilla, Colombia) under resolution number 3247 12th June of 2015. The funders had no role in study design, data collection and analysis, decision to publish, or preparation of the manuscript.

==============================
Background

With about 1,000 species in the Neotropics, the Eumaeini (Theclinae) are one of the most diverse butterfly tribes. Correct morphology-based identifications are challenging in many genera due to relatively little interspecific differences in wing patterns. Geographic infraspecific variation is sometimes more substantial than variation between species. In this paper we present a large DNA barcode dataset of South American Lycaenidae. We analyze how well DNA barcode BINs match morphologically delimited species.

Methods

We compare morphology-based species identifications with the clustering of molecular operational taxonomic units (MOTUs) delimitated by the RESL algorithm in BOLD, which assigns Barcode Index Numbers (BINs). We examine intra- and interspecific divergences for genera represented by at least four morphospecies. We discuss the existence of local barcode gaps in a genus by genus analysis. We also note differences in the percentage of species with barcode gaps in groups of lowland and high mountain genera.

Results

We identified 2,213 specimens and obtained 1,839 sequences of 512 species in 90 genera. Overall, the mean intraspecific divergence value of CO1 sequences was 1.20%, while the mean interspecific divergence between nearest congeneric neighbors was 4.89%, demonstrating the presence of a barcode gap. However, the gap seemed to disappear from the entire set when comparing the maximum intraspecific distance (8.40%) with the minimum interspecific distance (0.40%). Clear barcode gaps are present in many genera but absent in others. From the set of specimens that yielded COI fragment lengths of at least 650 bp, 75% of the a priori morphology-based identifications were unambiguously assigned to a single Barcode Index Number (BIN). However, after a taxonomic a posteriori review, the percentage of matched identifications rose to 85%. BIN splitting was observed for 17% of the species and BIN sharing for 9%. We found that genera that contain primarily lowland species show higher percentages of local barcode gaps and congruence between BINs and morphology than genera that contain exclusively high montane species. The divergence values to the nearest neighbors were significantly lower in high Andean species while the intra-specific divergence values were significantly lower in the lowland species. These results raise questions regarding the causes of observed low inter and high intraspecific genetic variation. We discuss incomplete lineage sorting and hybridization as most likely causes of this phenomenon, as the montane species concerned are relatively young and hybridization is probable. The release of our data set represents an essential baseline for a reference library for biological assessment studies of butterflies in mega diverse countries using modern high-throughput technologies an highlights the necessity of taxonomic revisions for various genera combining both molecular and morphological data.

Introduction

The ability to delimit and identify species is the foundation for addressing diversity issues in evolution, ecology, conservation, and biogeography. DNA barcodes potentially offer the opportunity for the rapid determination of species in large faunas, but reference libraries are needed to take advantage of this technique (Wirta et al., 2016; Hajibabaei et al., 2006). As of mid-2020, the Barcode of Life Database global repository (BOLD, http://www.boldsystems.org; Ratnasingham & Hebert, 2007) includes more than 9 million DNA barcode sequences for over 224,000 metazoans (700,000 BINs, including many not yet identified taxa) and 69,000 plant species. There are DNA barcodes from species in every country worldwide, with many supporting national barcoding initiatives. Each specimen in BOLD with a sequence longer than 500 bp is automatically assigned a global unique identifier (BIN, Barcode Index Number) based on the Refined Single Linkage (RESL) algorithm (Ratnasingham & Hebert, 2013). BIN assignments can be updated when new records reveal clear sequence divergence structure.

DNA barcodes accurately delimit species in a number of large-scale studies (e.g., birds, Hebert et al., 2004b; Kerr et al., 2007; moths, Hebert, DeWaard & Landry, 2010; Hausmann et al., 2011; Huemer et al., 2014; beetles, Hendrich et al., 2014; bees, Schmidt et al., 2015; dipterans, Morinière et al., 2019). They are often useful for discovering cryptic species, as has been shown with butterflies and flies (Hebert et al., 2004a; Smith et al., 2006; Van Velzen, Bakker & VanLoon, 2007; Riedel et al., 2013; Espinoza, Janzen & Hallwachs, 2017; Janzen et al., 2017; Dias et al., 2019; Tujuba, Hausmann & Sciarretta, 2020). In many cases, BINs correspond with traditional taxonomy. However, perfect congruence is rare (e.g., Hawlitschek et al., 2017; Pyrcz et al., 2018). While studies of the genetic diversity within a given species requires sampling from many localities (Bergsten et al., 2012), simple identification often requires only a single reference sequence (Hebert et al., 2003; Hausmann et al., 2013; Hawlitschek et al., 2017).

The utility of barcodes for describing several aspects of biodiversity depends on a strong correspondence between morphologically and genetically delimited entities. Although >20% of species pairs exhibit some level of incongruence in analyses at a continental scale (cf. Hausmann et al., 2013), the correlation increases significantly if the analyses are geographically restricted, such as a single country (Hausmann, 2011; Hausmann et al., 2013; Hendrich et al., 2014). For example, DNA barcodes accurately identified more than 95% of Argentine butterfly species (Lavinia et al., 2017). The success rate of DNA barcoding also varies among taxa, as can be seen among lepidopteran groups. Although some apparent differences among taxa may be due to biogeographic factors, DNA barcode species identifications were of more limited usefulness in neotropical Ithomiini butterflies (Elias et al., 2007) and Palearctic Elachistidae moths (Kaila & Stahls, 2006), but were more useful in the lepidopteran families Hesperiidae, Sphingidae, Saturniidae, Geometridae and Erebidae (Hajibabaei et al., 2006; Hausmann et al., 2011; Rougerie et al., 2014; Ortiz et al., 2017).

The primarily neotropical Eumaeini (Lycaenidae, Theclinae) contains more than a thousand species (Robbins, 2004) and represents one of the most rapid radiations among the butterflies. Taxonomic difficulties, external similarity, small size, rarity, high species richness, and restricted geographical distributions (at least of high montane species) are the most likely causes of the relatively scarce knowledge of this butterfly tribe. In contrast with other, better known families, lycaenids lack sufficiently illustrated identification keys, monographs, field guides, or checklists covering regions or countries in a comprehensive and updated manner. The use of DNA barcode sequences and BINs in this group has been limited, but congruence between morphology and barcode sequences is variable (Prieto, Micó & Galante, 2011; Faynel et al., 2011; Faynel, Busby & Robbins, 2012; Prieto et al., 2016; Cong et al., 2016, 2017; Prieto & Lorenc-Brudecka, 2017; Busby et al., 2017; Prieto, Nuñez & Hausmann, 2018; Faynel, 2019). In particular, in previous studies it appeared that strictly high Andean genera were more likely, on average, to show incongruence.

Incongruence between morphology and barcodes occurs when more than one BIN is detected in a traditionally recognized species or when a BIN number comprises members of more than one recognized species (Hebert et al., 2004a, 2004b). BIN discordance can be caused by unrecognized cryptic diversity whereas BIN sharing may indicate recently separated lineages that are still undergoing genetic differentiation. In both cases, an evidence-based taxonomic choice must be made, either to describe a new species (BIN split) or to synonymize two names (BIN sharing). These taxonomic decisions can increase the percentage of congruence between DNA and morphology-based analyses.

In this paper we present a large DNA barcode dataset of South American Lycaenidae. We analyze genus by genus how well DNA barcode BINs match morphologically delimited species. The general goal is to quantify the potential usefulness of reference libraries of DNA barcodes for identification and for resolution of taxonomic problems in this group. In previous studies (e.g., Prieto et al., 2016; Prieto, Nuñez & Hausmann, 2018; Faynel, 2019) we found that congruence between DNA and morphology varies among genera. We hypothesized that the incidence of congruence among strictly high Andean genera was lower than among lowland genera. A specific goal of this paper is to evaluate whether or not the ability of DNA barcodes to discriminate morphologically delimited species decreases in high elevation lineages.

Materials & methods

Morphology-based species identifications

The basis for identifying the species analyzed in this study is the checklist of Robbins (2004), which includes 1,058 species of Eumaeini in 83 genera. The checklist was updated using subsequent publications (e.g. Balint & Faynel, 2008; Prieto et al., 2008; Duarte & Robbins, 2010; Faynel et al., 2011; Prieto, 2011; Faynel, Busby & Robbins, 2012; Robbins, Heredia & Busby, 2015; Prieto et al., 2016; Prieto & Vargas, 2016; Busby et al., 2017). When necessary, identifications were verified through genitalic examination.

Sampling and sequencing

Collecting permits in Colombia were obtained from ANLA Agencia Nacional de Licencias Ambientales (00594 April 26th 2018). Tissue samples were taken from pinned Eumaeini (Theclinae) in the research collections of Carlos Prieto (RCCP) and Christophe Faynel (RCCF). We selected specimens collected in the past 10 years because older material is more likely to have degraded DNA. Samples came from Costa Rica, French Guiana, Colombia, Ecuador, Peru, and Brazil (Fig. 1). One to three legs were removed from each sampled specimen. The sample included 2,214 specimens of 541 species identified a priori based on the existing classification. The number of specimens per species ranged from 2 to 23.

Figure 1 Map of South America.

Distribution of sequenced material of Eumaeini (Lycaenidae, Theclinae).

DNA extraction, amplification, and sequencing of the COI barcode region were carried out by the Canadian Centre for DNA Barcoding (CCDB), Ontario, Canada, using standard high throughput protocols (Ivanova, DeWaard & Hebert, 2006; DeWaard et al., 2008). PCR amplification with a single pair of standard primers targeted a 658 bp region near the 5′ terminus of the mitochondrial cytochrome c oxidase I (COI) gene that included the standard 648 bp barcode region for the animal kingdom (Hebert et al., 2004a). Complete specimen data including images, voucher deposition, accession numbers, GPS coordinates, sequence and trace files are accessible in the Barcode of Life Data System (BOLD dataset: DS-CPCF Faynel-Prieto Neotropical Theclinae; doi: https://dx.doi.org/10.5883/DS-CPCF). Distance-based Neighbor joining (NJ), available on the BOLD website, was used to construct DNA barcode gene trees and to quantify sequence divergence. We analyzed the entire dataset and each genus with the NJ algorithm. In some cases, nearest neighbor genera with few species were combined in a single tree.

Congruence between morphology and BINs

BOLD currently contains close to 9,000,000 barcodes and over 700,000 BINs generated with the Refined Single Linkage (RESL) algorithm. RESL employs a three phased analysis to reach decisions on the number and circumscription of BINs (= MOTUs) in the sequence data set on BOLD (Ratnasingham & Hebert, 2013). It is much faster than other approaches, such as the generalized mixed Yule-coalescent model (Pons et al., 2006; Fujisawa & Barraclough, 2013), a critical requirement for the analysis of large data sets.

Morphological species were partitioned into three categories following the comparative methodology of Hausmann et al. (2013): (I) those in which there was a perfect match between morphological species and BINs; (II) splits: those were morphological species placed in more than one BIN and (III) merges: those where different species shared the same BIN assignment or mixtures where some individuals of a species shared a BIN with another morphological species. We re-examined each sample in the latter two cases by checking both the morphological identification and the alignment and trace files.

Barcode gaps

We analyzed barcode gaps to evaluate the hypothesis that incongruence between morphological species and BINs increase in high Andean lycaenid genera (e.g. Prieto, Nuñez & Hausmann, 2018; Faynel, 2019). The “barcode gap” is a comparison of intraspecific versus interspecific divergence among barcode sequences. A barcode gap exists if the intraspecific divergence (of a particular species) is smaller than its lowest interspecific divergence. For example, a small intraspecific divergence combined with a large interspecific divergence is a large gap. We compared these divergences in the entire dataset and in groups of genera partitioned by the elevational distribution of its species. The criterion for assigning elevational groups was that at least 90% of the species in a genus occur in (1) high mountain habitats (+2,200 m), (2) middle mountain habitats (1,220–2,200 m), (3) lowland habitats (0–1,200 m), (4) middle mountain + high mountain and, (5) or lowland + middle mountain.

As a quick visualization of barcode gaps, we made scatterplots showing maximum intraspecific variation plotted against the minimum distance to the nearest non-conspecific individual. A 1:1 relationship is the point at which the difference between the two is zero (Collins & Cruickshank, 2013). To determine sampling size bias, we also made scatterplots with the number of individuals in each species plotted against their maximum intraspecific variation. These analyses were performed for genera with at least two species and five sequences and for the groups of genera according to their elevational category. To evaluate if the divergence patterns for intraspecific variation and distances to nearest neighbor differ between the sets of species occurring in high mountain and lowland habitats, a Shapiro–Wilk normality test (Shapiro & Wilk, 1965), and a Kruskall–Wallis test were performed. The analyzes were carried out in R software.

Results

DNA barcode sequences at least 500 base pairs (bp) in length were successfully recovered from 1,839 specimens. These sequences were assigned to 556 BIN numbers that belong to 512 morphology-based species in 90 genera. From the congruence analysis (1,597 sequences, 558 BINs, 398 species, 52 genera) mean intraspecific variation ranged from a low of 0.1% in Paraspiculatus to a maximum value of 3.85% in Cyanophrys. Mean distances to nearest neighbor species ranged from 2.3% in Contrafacia to 10.4% in Aubergina. Altogether, 299 (75%) morphology-based species perfectly matched a unique BIN, while 36 species (9%) shared a BIN with up to six species, and 60 species (17.33%) were placed in two or more BINs. After reevaluating the morphology-based species based on the molecular results, congruence between morphology and BINs rose to 85%. However, BIN sharing and BIN splitting were particularly frequent in typically high montane genera such as Johnsonita, Rhamma, Podanotum, and Penaincisalia (Table 1).

Table 1 Summary of the percentages of congruence between BINs and morphology-based identifications.

Analysis for 52 genera of Eumaeini (Lycaenidae, Theclinae) represented by perfect matches, BIN splitting, and BIN sharing. Percentages were corrected (number in parentheses) when the BIN clustering indicated to the taxonomist a confirmed synonymy or cryptic species, in both cases we assume that the BIN designation was correct and the a priori morphological identification was wrong. Maximum intraspecific distance and minimum interspecific distances are highlighted in bold when a clear barcode gap exists. Some species can present BIN sharing and BIN splitting at the same time, which makes the sum of the percentages of perfect match, BIN sharing and BIN splitting exceed 100% for the genus.

Genus	% spp. perfect match	% spp. with two or more BIN	% spp. shared BIN	Mean Intra dist. % Normalized	Max intra dist.%	Min inter dist. %	Mean inter dist. %	Max inter dist. %	n. species	n. sequences	
(Option 1)	(Option 2)	(Option 3)						398	1,597	
Paraspiculatus	78 (78)	0 (0)	22 (22)	0.11	0.31	0.98	4.25	6.57	9	13	
Brangas	86 (100)	14 (0)	0 (0)	0.53	2.83	1.55	5.36	8.8	7	41	
Thaeides	0 (100)	50 (0)	50 (0)	1.31	3.79	0.31	7.66	9.66	4	18	
Enos	100 (100)	0 (0)	0 (0)	0	0	6.05	6.05	6.05	2	5	
Evenus	44 (44)	22 (22)	33 (33)	0.68	4.77	1.86	5.1	7.59	9	24	
Atlides	72 (82)	27 (18)	0 (0)	0.65	3.63	3.3	6.15	9.25	11	71	
Arcas	100 (100)	0 (0)	0 (0)	0.08	0.16	3.91	4.28	4.59	3	5	
Theritas	69 (92)	31 (8)	0 (0)	1.15	4.47	2.34	6.15	9.72	13	78	
Johnsonita	40 (80)	40 (0)	40 (40)	3.35	8.6	0	4.76	8.98	5	28	
Brevianta	100 (100)	0 (0)	0 (0)	0.6	1.68	3.13	4.02	4.58	4	17	
Micandra	83 (83)	17 (17)	0 (0)	0.29	1.88	1.78	4.46	6.4	6	67	
Rhamma	26 (47)	40 (20)	53 (53)	0.96	5.93	0	4.17	7.92	15	190	
Timaeta	100 (100)	0 (0)	0 (0)	0.55	3.87	4.09	5.04	8.31	6	18	
Penaincisalia	68 (77)	23(13)	9 (9)	0.83	5.52	0.46	5.15	8.75	21	123	
Lathecla	100 (100)	0 (0)	0 (0)	0.49	2.04	3.46	4.82	8.11	3	7	
Podanotum	67 (89)	33 (11)	0 (0)	0.9	4	1.63	3.79	6.17	9	64	
Thereus	67 (76)	14 (5)	19 (19)	1.26	8.45	0	6.93	12.04	21	50	
Rekoa	67 (100)	33 (0)	0 (0)	1.56	2.81	6.89	7.47	8.06	3	7	
Arawacus	80 (80)	20 (20)	0 (0)	0.69	3.16	2.81	4.62	10.7	5	19	
Contrafacia	100 (100)	0 (0)	0 (0)	0.13	0.35	1.94	2.34	2.49	2	7	
Kolana	67 (100)	33 (0)	0 (0)	0.04	0.15	4.42	5.52	6.24	3	8	
Ocaria	75 (87)	25 (13)	0 (0)	0.4	2.76	2.65	4.9	8.95	8	35	
Cyanophrys	72 (100)	28 (0)	0 (0)	1.01	5.82	2.31	4.4	7.37	7	26	
Thestius	100 (100)	0 (0)	0 (0)	0.05	0.15	4.9	6.23	7.05	3	6	
Allosmaitia	100 (100)	0 (0)	0 (0)	0.91	1.86	6.08	6.66	7.4	2	8	
Janthecla	60 (60)	0 (0)	40 (0)	0.32	0.93	0.77	6.49	9.74	5	10	
Lamprospilus	83 (92)	17 (8)	0 (0)	0.73	4.79	3.46	5.8	8.49	12	51	
Arzecla	75 (100)	25 (0)	0 (0)	0.05	0.31	3.78	6.78	8.06	8	43	
Arumecla	100 (100)	0 (0)	0 (0)	0.04	0.16	5.91	6.66	7.78	5	11	
Electrostrymon	80 (100)	20 (0)	0 (0)	0.68	2.89	1.55	6.02	7.16	5	22	
Strymon	94 (94)	6 (6)	0 (0)	0.41	3.09	2.67	8.11	12.21	17	27	
Tmolus	60 (100)	40 (0)	0 (0)	0.44	0.93	2.65	4.59	8.63	5	12	
Nicolaea	100 (100)	0 (0)	0 (0)	0.3	1.08	2.18	7.07	10.56	11	18	
Ministrymon	83 (83)	17 (17)	0 (0)	1.01	2.82	1.55	5.32	7.06	6	11	
Gargina	75 (75)	25 (25)	0 (0)	0.75	8.96	4.27	7.75	10.01	4	21	
Siderus	71 (100)	29 (0)	0 (0)	2.24	5.83	0	7.4	10.9	7	20	
Theclopsis	75 (75)	25 (25)	0 (0)	1.41	2.51	4.41	6.12	7.62	4	7	
Ostrinotes	100 (100)	0 (0)	0 (0)	0.26	0.82	2.2	6.2	8.42	7	13	
Strephonota	65 (77)	27 (15)	11 (11)	0.85	7.99	0	4.82	10.39	26	103	
Panthiades	88 (88)	12 (12)	0 (0)	0.76	3.34	1.87	4.53	6.55	8	20	
Oenomaus	61 (61)	26 (26)	26 (26)	1.17	7.63	0	5.91	9.9	23	86	
Porthecla	100 (100)	0 (0)	0 (0)	0.49	1.55	3.45	5.59	7.26	5	13	
Thepytus	67 (67)	33 (33)	0 (0)	0.61	2.02	4.73	5.81	7.61	3	8	
Parrahasius	100 (100)	0 (0)	0 (0)	0.07	0.31	2.6	3.21	3.77	3	7	
Michaelus	100 (100)	0 (0)	0 (0)	0.08	0.15	4.41	4.99	5.43	4	7	
Ignata	100 (100)	0 (0)	0 (0)	0.19	0.86	2.05	4.27	7	5	9	
Olynthus	71 (71)	0 (0)	29 (29)	0.2	0.81	0.65	2.67	4.57	14	36	
Marachina	100 (100)	0 (0)	0 (0)	0.31	0.93	5.88	6.05	6.22	2	5	
Aubergina	100 (100)	0 (0)	0 (0)	0.17	0.32	9.99	10.36	10.68	2	13	
Iaspis	100 (100)	0 (0)	0 (0)	0.32	1.44	2.58	4.56	6.23	5	9	
Erora	92 (100)	8 (0)	0 (0)	1.3	4.92	2.02	6.06	8.9	12	23	
Chalybs	89 (100)	11 (0)	0 (0)	0.22	1.17	0.34	5.89	8.76	9	57	

Barcode gaps

The percentage of species with a barcode gap in the complete data set was 87.2%. However, the proportion of species from habitats at different elevations in the datasets affected barcode gap frequency. Gaps were observed in 95.7% of the species from lowland ecosystems (0–1,200 m) while only 61.7% of species from exclusively high montane ecosystems (>2,200 m) had clear barcode gaps (Figs. 2, 3). The trend towards a higher percentage of barcode gaps in lowland species was also found when including genera with species distributed in both lowlands and mid-montane habitats (89.5%), exclusively mid montane habitats (82.8%) and genera with exclusively mid- and high-montane species (73.8%) (Table 2). The divergence values to the nearest neighbor were significantly lower in the high Andean species (H = 26.6, p-value = 0.0000002436), while the intra-specific divergence values were significantly lower in the lowland species (H = 6,49, p-value = 0.01084).

Figure 2 Distance data for the barcode gap analysis.

Scatterplots are provided to confirm the existence and magnitude of barcode gaps for the complete set of species and for groups of genera including mid + high mountain and lowland + mid-mountain species. The first two scatterplots show the overlap of the max and mean intraspecific distances vs the interspecific (nearest neighbour) distances. In the three groups of altitude most species fall above the 1:1 line, indicating the presence of a barcode gap (for percentages see Table 2). The third scatterplot plots the number of individuals in each species against their max intraspecific distances, as a test for sampling bias.

Figure 3 Distance data for the barcode gap analysis.

Scatterplots are provided to confirm the existence and magnitude of barcode gaps for exclusively high mountain genera, mid-mountain genera and lowland genera. The first two scatterplots show the overlap of the max and mean intraspecific distances vs the interspecific (nearest neighbor) distances. Lowland genera show a higher percentage of species with local barcode gaps (points above the 1:1 line) than mid and high mountain genera (see Table 2). The third scatterplot plots the number of individuals in each species against their max intraspecific distances, as a test for sampling bias.

Table 2 Percentages of congruence and barcode gaps.

Percentage of species with barcode gap and percentage of species with perfect congruence between BINs and morphospecies for each group of genera depending on altitude. Exclusively lowland genera present a higher percentage of species with barcode gaps than exclusively high mountain and mid-mountain species.

	Individuals	Genera	Species	BINs	% spp with barcode gap	% spp with perfect congruence BIN vs morphology	
Complete set	1,834	89	485	556	87.2	84.6	
Mid-mountain + high mountain	741	11	84	112	73.8	79.1	
Lowland + mid-mountain	1,078	25	244	261	89.7	86.0	
High mountain	482	4	47	67	61.7	71.4	
Mid-mountain	259	7	35	45	82.8	82.2	
Lowland	339	14	118	114	95.7	85.2	

Discussion

BIN sharing

In this study, we obtained 1,839 sequences of 512 species distributed in 90 genera for 557 BIN numbers, representing 78% of the available data on BOLD for the Eumaeini. From the set of specimens that yielded COI fragment lengths of at least 650 bp, 75% of the a priori morphology-based identifications were unambiguously assigned to a single Barcode Index Number (BIN). After a taxonomic a posteriori review, the percentage of perfect matching rose to 85%. Very low levels of interspecific barcode variation can reflect overlooked synonymy if misidentifications are ruled out (e.g. Puillandre et al., 2011), but low genetic divergence, particularly based on just one genetic locus, does not automatically invalidate established taxonomy. In cases of recent phylogenetic divergence, phenotypic differentiation can occur more rapidly than the complete sorting of mtDNA into the new, separated lineages. The decision to consider two species names as synonyms must be made by a taxonomist. That is why in our study we strived for accurate identification before and after delimiting the species using molecular data. When species pairs with low barcode divergences are recovered as monophyletic groups in the cladograms or identification trees, and their morphology is highly divergent, they can be validated as two different species, particularly if they are sympatric. There is no fixed threshold level of divergence that indicates species status because the percentage of divergence that would indicate whether two entities belong to the same species depends on the taxonomic group being studied and its evolutionary history. Nevertheless, most studies have found that COI divergences rarely exceed 2% within named and morphologically validated species, while members of different species typically show higher divergences, and it has been shown repeatedly that this ‘threshold’ can be used in many or most metazoans to determine species status (Ratnasingham & Hebert, 2007, 2013). However, distance and geographic isolation are two aspects that must be taken into account when delimiting biological entities based on established thresholds. Two entities living in sympatry can be considered different species even when there are small genetic divergences of 2% or less. But if these same entities are geographically distant, a 2% divergence may be considered irrelevant to define them as separate species.

The percentage of clearly different morphological species grouped within the same BIN was predominantly high for the montane genus Rhamma, where the BIN BOLD:ABX0547 is shared by five well-differentiated morphospecies, most of them flying in sympatry. We suggest that most of the cases of BIN sharing between morphologically divergent high mountain species represent recently separated lineages that are still undergoing genetic differentiation. As most of these cases were recovered as monophyletic clades in the trees, a lower basic threshold setting in the algorithm would separate these species into different BIN numbers. However, it should be noted that with such modified settings and parameters the number of cases of BIN discordance in the same group of species may increase. In the case of the genus Rhamma the assignment of a single BIN number for clearly different morphological species is a result of the basic settings chosen for delimitation of sequences into BINs.

Incomplete lineage sorting is relatively common in recently and rapidly radiating species groups as these species often have not yet had the necessary time to fix alternative haplotypes or alleles (Galtier & Daubin, 2008). As a result, the relationships of incipient species typically progress from initial polyphyly through paraphyly and reach monophyly once lineage sorting is complete in the two sister species. Thus, in mtDNA analyses, relatively young species may appear polyphyletic or paraphyletic owing to incomplete lineage sorting (Tang et al., 2012). This phenomenon seems to be particularly common in high Andean genera such as Rhamma and has important effects on species identification and delimitation based on genetic sequence analysis. Further studies comparing genetic distances of sympatric and allopatric populations of several pairs of species can help to detect evidence of incomplete lineage sorting, and its prevalence in high mountain species of Theclinae.

BIN splits

High levels of intraspecific barcode variation often reflect cryptic species (e.g. Puillandre et al., 2011, 2012). However, deep barcode splits can also be the result of the recovery of pseudogenes, as a consequence of hybridization, or Wolbachia infection (Werren, Baldo & Clark, 2008; Huemer et al., 2018; Mally, Huemer & Nuss, 2018). High percentages of BIN splits were found in some genera with typical mid- and high mountain representatives such as Podanotum, Johnsonita, Thaeides and Rhamma.

As noted by Prieto, Nuñez & Hausmann (2018), the genus Rhamma includes several species presenting both types of discordance, BIN sharing and BIN splitting (e.g., R. arria and R. bilix). Species with a wider geographical distribution in high Andean ecosystems, seem to show a greater number of incongruences. Morphologically identified specimens were placed in three well-differentiated BINs for R. arria (BOLD:ABX0547, BOLD:ADD3784, BOLD:ADD3785) and four BINs for R. bilix (BOLD:ACF3699, BOLD:ABX0491, BOLD:ADD1839, BOLD:ABX0493). These clades might correspond to either divergent conspecific lineages, or unconfirmed putative species separated, in some cases, by deep genetic divergences. Cases of mitochondrial introgression can hinder the delimitation of some Eumaeini species in the genus Calycopis (Cong et al., 2017), and we suppose that such processes occur more frequently in high Andean genera. Introgressive hybridization may have been common throughout the evolutionary history of these genera which are, therefore, of particular interest to taxonomists and evolutionary biologists because partial and unequal gene exchange can have important effects on the dynamics of speciation and phylogenetic patterns (Grant, 1998; Grant, Grant & Petren, 2005; Funk & Omland, 2003), and affect species identification and delimitation based on DNA sequences.

Barcode gap analysis

A useful display of distance data for species delimitation is a scatterplot showing for each species the maximum intraspecific variation against the minimum distance to the nearest non-conspecific species (‘nearest neighbor’), with a 1:1 slope representing the point at which the difference between the two is zero (Collins & Cruickshank, 2013). This type of representation shows the barcode gap for each species in the dataset and can be an accurate display of the percentage of species in the study group that have a barcode gap (Figs. 2, 3). Since the identification of species and the delimitation of taxonomic entities using barcodes, depend on the existence of a clear DNA barcode gap, a quick visualization of the existence of these gaps in each species can indicate the usefulness of the DNA barcoding approach in a given genus.

Major topographic and climatic variations are important factors that determine the nature of South American biodiversity. The geography of South America, together with its climate and biodiversity, have evolved over a very long period, initiated about 100 MY ago on the ancient Gondwanan continent, although the Andes rose much later (about 25 MY ago). Many species of butterflies are found near, or within, the complex valley systems of the Andes, a result of the combination of at least two important factors: altitudinal gradients and geographic barriers created by the intricate system of valleys and ridges (e.g. Holzinger & Holzinger, 1994; Willmott, 2003; Ebel et al., 2015). The tectonic rise of the Andes has created new environments and modified others, and the uplift of the cordilleras has separated butterfly communities favoring the evolution of allopatric vicariants. Dramatic changes in global climate during glaciations, accompanied by major adjustments in vegetation, created new biomes which again may have stimulated the evolution of new species and subspecies, specially at high altitudes (Purser, 2015; Pyrcz et al., 2017). These changes are quite rapid on a geological scale, and certain subspecies, notably among the high altitude pronophilines (Satyrinae), seem to have evolved since the last glacial maximum 20,000 years ago (Adams, 1985; Pyrcz, Wojtusiak & Garlacz, 2009; Casner & Pyrcz, 2010). Several phylogenetic studies of butterflies indicate that the most recent diversification events tend to occur at high elevations and that the highest altitude species and subspecies are the youngest (e.g., Jiggins et al., 2006; Casner & Pyrcz, 2010; Pyrcz et al., 2017).

Due to incomplete lineage sorting in very young species, it is not easy to accurately define taxonomic boundaries, and additional difficulties may be caused by hybridization. Although the RESL algorithm as the basis for the BIN system (Ratnasingham & Hebert, 2013) provides a powerful tool to propose primary, tentative species hypotheses for large datasets (Ratnasingham & Hebert, 2007), such an mtDNA-based approach cannot prove the absence of gene flow and still depends on arbitrary, a priori settings and assumptions. The efficiency of these methods largely depends on the accumulation of mtDNA mutations since species separation, and thus can only delimit lineages with sufficiently long isolation (Rannala, 2015). Moreover, nothing is known about how the kind of speciation process (vicariance of an existing species versus a small founder colonization) might affect the ability of barcodes to identify species correctly. We assume that incomplete lineage sorting and occasional hybridization are usual phenomena in the very young species of South American Eumaeini, and that these are the two most likely causes of the low percentages of DNA barcode gaps found in high Andean species in comparison with the older species from the lowlands.

However, other very species-rich groups of Andean butterflies with recent speciation processes such as the subtribe Pronophilina, have shown very high percentages of barcode gaps and perfect congruence between morphology and DNA barcodes (e.g. Marín et al., 2017; Pyrcz et al., 2018). This shows that, in certain groups, other biological factors allow young high Andean species to present more complete DNA lineage sorting in short periods of time. In the case of Pronophilina, the low vagility of its species could be a determining factor that limits gene flow between separate populations and promotes lineage sorting.

Conclusions

In mtDNA analyses, relatively young species may appear polyphyletic or paraphyletic owing to incomplete lineage sorting, and other aspects such as introgressive hybridization may have been common throughout the evolutionary history of Eumaeini genera. Partial gene exchange can have important effects on the dynamics of speciation, and affect species delimitation based on DNA sequences. These phenomena seem to be particularly common in high Andean genera such as Rhamma and to have important effects on species identification based on genetic sequence analysis.

Since we found evidence, at least in the tribe Eumaeini, that relatively young species in young ecosystems tend to have more incongruences between morphology and DNA delimitation, and thus lower percentages of DNA barcode gaps, it would be interesting to find out if there are similar patterns when comparing groups of species belonging to related genera in young and old ecosystems at the same altitude. This could be done by comparing a group of high mountain species from the northern part of the Andes in Venezuela, with their relatives in the central part of the Andes in Peru. These two regions exhibit a different geological age of around 40 MY, with the Venezuelan part being the youngest.

Supplemental Information

Supplemental Information 1 Sequences of eumaeini from South America.

Click here for additional data file.

For help and support in many ways we acknowledge Daniel Augusto Mantilla, (Popayán, Colombia), Evgeny V. Zakharov (CCBD, Guelph, Canada), Pierre Boyer (France), Gregory Nielsen, (Colombia), Myriam Nicolle Pertuz (Barranquilla, Colombia), Tomasz Pyrcz (Kracow, Poland), Michael Balke (ZSM, Munich, Germany), Soranggy Cruzco (Barranquilla, Colombia) and Zsolt Balint (HNHM, Budapest, Hungary).

Additional Information and Declarations

Competing Interests

Author Contributions

Field Study Permissions

Data Availability

The authors declare that they have no competing interests.

Carlos Prieto conceived and designed the experiments, performed the experiments, analyzed the data, prepared figures and/or tables, authored or reviewed drafts of the paper, and approved the final draft.

Christophe Faynel conceived and designed the experiments, performed the experiments, analyzed the data, prepared figures and/or tables, and approved the final draft.

Robert Robbins analyzed the data, authored or reviewed drafts of the paper, and approved the final draft.

Axel Hausmann analyzed the data, prepared figures and/or tables, authored or reviewed drafts of the paper, and approved the final draft.

The following information was supplied relating to field study approvals (i.e., approving body and any reference numbers):

Collecting permits were obtainend from ANLA Agencia Nacional de Licencias Ambientales de Colombia 00594 (April 26th 2018)

The following information was supplied regarding data availability:

The raw data are available at BOLD systems: DS-CPCf (http://www.boldsystems.org/index.php/Public_SearchTerms).

All sequences are available in the Supplemental File.

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
