# Peer review of "Congruence between morphology-based species and Barcode Index Numbers (BINs) in Neotropical Eumaeini (Lycaenidae)"

_PeerJ, doi:10.7717/peerj.11843_

## Round 0.1 · original submission · Major Revisions

Three reviewers have now assessed your manuscript. While they all agree in that the study is sound and presents interesting findings, a series of major concerns have been raised, including the lack of statistics supporting the conclusions drawn from the study, the suggestion to expand the analyses to explore species delimitation, or the necessity of including further information in some sections. Please read these comments carefully and take them into consideration while preparing your revised version. In addition, please make sure that the manuscript is checked by a native English speaker.

Looking forward to reading the revised version of your manuscript!

Best regards,

Rosa Fernández

Reviewer 1 ·

Basic reporting

-I recommend using the same format in bibliographic references, sometimes some have the full name of the journal, sometimes not.
- In Table 2, second column says ‘Indivuduals’ it should says ‘Individuals’
-Please, it´s necessary to verify some values and links, more details in the manuscript.
- There some spelling errors that are marked in the manuscript.

Experimental design

No comment

Validity of the findings

No comment

Additional comments

The manuscript is worth publishing.

Annotated reviews are not available for download in order to protect the identity of reviewers who chose to remain anonymous.

Reviewer 2 ·

Basic reporting

Overall, this paper is organized in a clear way. While I am not a native English speaker, I noticed that English is not perfect throughout, though the problems seem to be mostly minor ones. Particularly in the discussion, there are several claims about previous findings without references (e.g. sentences ending on lines 210, 225 and 250). In general, and again particularly in the discussion, the used literature is not extensive enough and many important papers have not been considered. The figure captions are not very clear and well-written (e.g. the sentence "Each point represent the collecting site of each individual" in Fig. 1 caption). The figure 2 and 3 caption start by saying "Three scatterplots are provided..." but there are nine in each.

Experimental design

The paper publishes an important and remarkable dataset of South American butterflies and clearly merits becoming published. The set research questions are relevant and meaningful, although the used methods are not particularly sophisticated. For example, the authors are concluding things based on visual patterns seen in the scatterplots, but their observations are very difficult to verify and as the differences were not shown to be statistically true, it is not easy to confirm if the patters actually even exist. It would not have been difficult for example to statistically (ANOVA etc.) investigate if the divergence patters (intraspecific variation, distances to NN) differ between the sets of species occurring in different altitudes. It could even be studied statistically (regression) if altitude has an effect to the mentioned patterns of divergence. I find the lack of statistics a clear weakness in this paper as, indeed, it is not clear if the "observed" patterns really hold true. The methods are described mostly in a sufficient manner. The authors used mostly standard analytical tools of BOLD that is not wrong, but some fine-tuning of the figures would not hurt (for example, due to a bug in the BOLD tools there is not space after the words "Dist"/"NN" and "(%)".

Validity of the findings

As stated above, the generated data is valuable and extensive. The data is made publicly available. I would find the overall logic of the discussion the weakest part of this manuscript and would like to elaborate this opinion further. First, the authors conclude (from line 335) that "relatively young species in young ecosystems tend to have more incongruences...". Actually, this represents circular reasoning. The thing that the species are young is based only a low sequence divergences. No more rigorous evidence for the young age is presented, although indirectly this is supported also by the relatively young age of the ecosystems they are inhabiting. Therefore, if young species are defined by small divergences, it is hardly surprising that young species are having small divergences. What else it could be, because it is so by definition? Second, what do the author mean by young age of species? All species have equally long evolutionary history. A species may turn to another species without a split of the lineage, or they may "technically" become new species as a result of such a split. Exactly, when do the authors think the new species was born? Is there evidence that the mutation rate of different lineages keeps constant? This question is further complicated by the fact that intrinsically (this is not discussed) it seems evident that the authors use different criteria for "species" under different connections. As they are discussing the hybridization and gene flow in several occasions, it seems that they think species being more or less reproductively isolated entities. But then they are telling that montane species are typically allopatric. How much do we know if the allopatric populations/species are capable to reproduce with each other? I feel that the authors are deep in the swamp what comes to the age of the ages of species and how they decipher the term species. Additionally, they seem to have adopted an opinion that they "know" where the true species limits lie. This is apparent also in the other sections of the paper as the functionality of DNA barcoding is compared to the prevailing taxonomy, but it is not discussed at all how accurate/inaccurate this taxonomic framework could be. Maybe there are false species included. Maybe some of the allopatric populations on montane species are not valid species at all (whatever it means as delimitation of allopatric populations is inherently arbitrary). I find that the discussion does not consider factors like these to the sufficiently degree. It is certainly tempting to rely on the idea that clear boundaries between species really exist, but this is hardly true. By this I do not mean that these matters should not be discussed at all. My point is that I find the discussion not very in-depth and not considering all relevant aspects. Third, the introduction sets some hypotheses that however are not explained and lack references. Especially it should be explained why the authors want to study the hypothesis (line 118) that the power of DNA barcodes decreases along with the altitude. This is discussed, weirdly, in more detail in material and methods, in lines 166-180. I find that this section would fit much better to the introduction. However, the reasoning in this section lacks any references. They also refer to their own previous studies (line 166) but do not give a reference.

Additional comments

Despite some criticism, I find the data and findings of this paper valuable and I am happy to recommend it after moderate revisions. Besides my comments above, I would like to raise some smaller issues:
Line 156: please clarify "outcome is deterministic". I don't understand what this exactly means.
Line 556: Although not many individuals per species was analyzed, the authors found a remarkable number of intraspecific BIN splits. It would have been nice to see this assessed by statistical means. How many splits could be expected if all species were sampled by, say 20 individuals? This value is naturally an underestimation, but this is not discussed.
Line 218: "...it is most likely that they can be validated as two different species". I do not find such things as questions of likelihood. It is more a question of how we define the term "species". I would leave the likelihood thing out and just write. "...is highly divergent, they can be validated as two different species, particularly is both entities...".
Line 251-252: Wolbachia does not itself result in barcode splits. It may mediate spread of introgressed mitochondria in a new species. These bacteria do not cause mutations but may greatly affect to the allele frequencies.
Lines 264-265: "...we suppose that such processes in high Andean species occur more frequently". This requires elaboration. Why do you think this being the case?
Line 268: "Phylogenetic results" is a bit weirdly stated. Perhaps "phylogenetic patterns"
Line 279-280: Here you give an idea that DNA taxonomy is about use of DNA barcodes in species delimitation that is not true. DNA taxonomy desires using multiple genes in species delimitation.
Lines 300-301: What is the true evidence that the species have evolved since the last glacial maximum? Is this based just on low genetic divergences?
Lines 309-310: All algorithmic species delimitation methods are based on arbitrary a priori settings and assumptions. It is exactly because of this what makes delimitation of species inherently arbitrary, especially with allopatric populations.
Line 314: This sentence includes an idea that a single the "true" taxonomy really exists.
Line 322: How can you know that in this case this happened in short period of time? It is a widespread problem in this manuscript that somehow, magically, you "know" how old different species are.

Small grammatic issues:
Line 87: The word "Lepidopteran" should not be capitalized'
Line 227: italicize "Rhamma"
Line 284: Remove and extra comma (not sure if this sentence makes good sense, by the way; certainly all continents have a long geographic history)
Line 292: intrincated > intricated

Reference section is with many inconsistencies and small glitches. Just in the first line (359) there are two such glitches (extra "s” and a lacking space). Some journals are abbreviated, some not. Line 374 gives "pp." for a journal article. After the volume number, both comma and colon are used. Line 448 is with species name without a space. Line 468 lacks the volume and page range. It varies after how many authors "et al." is used in the authors list. Lines 499 and 500 are both with an extra space. Line 553 is with two words together. Lines 570 and 571 are with extra hyphens. Some scientific names in article titles are italicized, some not. Just to mention some examples.

·

Basic reporting

The manuscript is overall clean and clearly written, but some light editing is needed throughout, e.g., "These phenomena seem to be particularly common in high Andean genera such as Rhamma and have important effects on species identification based on genetic sequence analysis" - compare with the original sentence.

I suggest that some sections of the manuscript could be improved by clarifying some issues and providing greater background information and references to relevant literature, especially more recent studies. The Introduction could clarify two of the main goals of developing DNA barcode libraries and go on to say why both of these are relevant to the study group. More information could be presented about the background of the study group. And, throughout the manuscript, there are a number of studies, especially published in the last 10-15 years, that would be relevant to cite in relation to Neotropical butterfly biogeography, e.g. on various groups of Melitaeini, Ithomiini, Adelpha, Heliconius, Pronophilina, Papilionidae, Riodinidae.

The authors are to be commended on their thorough approach to making all relevant data publicly available through BOLD.

Experimental design

Overall, the methods used and the data used are very well described and justified, with the exception perhaps of a little more detail about the parameters in RESL analysis, and why alternative delimitation approaches were not used could be better justified.

As mentioned above, the knowledge gap being filled (existing library of eumaeine sequences and understanding of eumaeine taxonomy) could benefit from greater clarification in the introduction. I also suggest that the authors consider conducting RESL analyses for subsets of their data, e.g. split into 'lowland' and 'montane', to see whether methodological issues such as trying to fit parameters to a highly variable dataset are reducing the effectiveness of the approach in montane groups. Furthermore, an analysis of the extent to which failures of the analysis to discriminate species result from low divergence between monophyletic groups vs parar/polyphyly would be very useful in directing subsequent discussions about the causes of these analytical failures. Finally, and throughout the manuscript, there should be a constant awareness about the importance of geography in explaining barcode divergence and affecting the apparent success or failure of species delimitation, both in this and other studies.

Validity of the findings

This is a massive and very significant dataset for a highly complex and understudied group, yet I think the authors could still better emphasize and quantify this contribution.

The conclusions are largely sound, except a bit qualitative, and there is a strong emphasis on discussing recent diversification and incomplete lineage sorting, whereas it is not clear how many cases of failed delimitation are due to these issues. The discussions of Andean butterfly biogeography are sometimes a bit generalized and lacking in references to recent publications, as mentioned above.

Additional comments

Overall, this is a really remarkable study which will provides a huge dataset of great value to Neotropical butterfly researchers. I made some more specific comments below.

Background
I am not sure infraspecific variation in Lycaenidae can be considered 'great', in comparison with many Nymphalidae, for example. The significant point is that geographic infraspecific variation can appear more substantial than variation between species, which is often very slight.

Methods
It seems inaccurate to say that it is the BIN system that delimitates MOTUs, rather MOTUs are delimitated by the RESL algorithm and each is labeled with a Barcode Index Number (BIN).

L 57: 'enumeration' is a bit vague, DNA barcodes can provide an estimate of how many species are in a sample without needing to put names on any of them (and hence without needing a reference library).

L 73: Elias et al 2007 is not a very appropriate reference here, instead it would be better placed on line 76 as a counter-example of the usefulness of barcodes. Better examples from Lepidoptera, in which descriptions of taxa were actually facilitated by DNA barcodes, can be found, such as Dias et al. (2019, "DNA barcodes uncover hidden taxonomic diversity behind the variable wing patterns in the Neotropical butterfly genus Zaretis (Lepidoptera: Nymphalidae: Charaxinae)") or Espinoza et al. (2017, "17 new species hiding in 10 long-named gaudy tropical moths (Lepidoptera: Erebidae, Arctiinae").

L 80: "The usefulness of DNA barcodes for describing biodiversity patterns depends on a strong correlation between morphologically and genetically delimited species" – again, this is a bit unclear. DNA barcodes have two common and distinct applications – 1) as a supplementary set of characters alongside morphology, ecology, other DNA markers, or anything else that can be used, to help delimit species and build a classification; 2) as a tool for identifying unknown samples by matching them to a library of identified sequences. These two points are mentioned in L 115-116, but I think they could be made more clearly at the start of the Introduction, to lead on to which of these aspects, or both (as is likely), will benefit from this study and dataset.

L 94-96 need a bit more elaboration for those not familiar with the group; what is meant, for example, by 'hard sampling' or 'restricted geographic distribution'? A few sentences about their biology might also be nice, e.g. behavior, hostplants, longevity, broad geographic patterns of community diversity etc. Furthermore, a couple of sentences about the state of Neotropical eumaeine taxonomy would be useful too – how many undescribed species are known, how frequently are new undescribed species uncovered, what are the principal morphological characters that have been used to build the current classification, are there large genera so in need of complete revision that their current classification is really rather tentative?

L 109-110: Or, in many cases, the decision is to gather further information, such as additional genetic markers or add samples to fill in geographic variation, to help resolve the situation.

L 117-118 (and L 167-169): this hypothesis presumably refers back to the empirical observation mentioned in L 102-103, but if this is an important goal of the paper it could merit a bit more discussion, e.g. theoretically why might high elevation species be less easy to delimit/identify? Lower gene flow because of linear ranges and fragmented populations, and therefore greater infraspecific geographic variation in barcodes? More recent speciation because these habitats are geologically younger? Habitat filtering which tends to lead to less phylogenetically diverse communities at higher elevations (e.g., Chazot et al. 2014, "Mutualistic Mimicry and Filtering by Altitude Shape the Structure of Andean Butterfly Communities")?

L 126-127: would seem more accurate to me here to say identification of samples, rather than species.

L 136: 541 species identified a priori based on the existing classification, presumably?

L 145: Whenever I mention 'Neighbor-joining' it brings down the wrath of most reviewers, so some justification as to why you used this method to construct trees (vs a ML or Bayesian approach, for example) would be good. What nucleotide substitution models were used, and what about other kinds of settings (e.g., dealing with missing data)?

L 155-156: what is beneficial about a 'deterministic' method, and what does that mean anyway? Also, being faster is not necessarily a reason for using a method, what other reasons support use of the RESL algorithm over other species delimitation approaches, e.g. GMYC, PTP?

L 185: not clear why the 'congruence analysis' contains only 1597 of the 1839 x >500 bp sequences obtained.

L 192-195: This is unclear, how can species that 'cannot be discriminated by DNA barcodes' have 'diagnostic barcodes'? Furthermore, what is this 'taxonomic decision', presumably you did additional genitalic dissections or analysis of other characters and found consistent differences between samples that coincided with DNA barcode differences, supporting the need to re-evaluate species limits? This retroactive revision should be described in the methods.

L 198: I may have missed this in the Methods, but what does this percentage indicate? How exactly was it determined whether an individual species 'had' a barcode gap? Ah, I see it is described rather well in L 273-278, I suggest moving this section to the Methods.

L 209: I'd suggest opening the Discussion with a summary statement about the main outcomes of the study, e.g. 'In this study we obtained X new DNA barcodes for Y species of Neotropical Eumaeini, increasing by Z% the number of species for which there are barcodes now available in this group. We found such divergence corresponding to etc etc.' Then move on to the details.

L 214-215: this is a bit of a woolly statement, who is to determine whether a particular taxonomist is 'competent'?

L 219: 'particularly', rather than 'mainly'

L 222-225: some references to support these assertions would also be useful. Furthermore, some discussion about the critical aspect of geographic distance between samples would be useful. A 2% divergence between samples from the same site is likely significant, but a 2% divergence between samples from Mexico and SE Brazil is almost meaningless.

L 228: sympatric morphospecies, or allopatric? If allopatric, their 'species' status is subjective.

L 234-236: Would it be worth exploring analyses of smaller subsets of the data, e.g. if you repeated the analysis for just 'montane' genera, would different parameters be selected and a better agreement between MOTUs and traditional species be found?

L 237: "A relatively small number of cases of BIN sharing represent paraphyly or polyphyly that could result from incomplete lineage sorting…", some introductory statement is needed to tie this to the previous paragraph.

L 250-253: Also, critically, as mentioned above, intraspecific divergence can reflect geographic separation, e.g. 6% divergence in Amiga (formerly Chloreuptychia) arnaca between eastern and western Ecuador.

L 256-257: Presumably, you mean conspecific sequences from distant localities were more likely to be divergent, but this sentence could also mean that DNA divergence between samples of species 1 from sites A and B is greater than in species 2 also occurring in sites A and B, when overall species 1 has a broader geographic distribution than 2. So, some clarification is needed.

L 259-260: again, do these BINs correspond to geographic clusters, or do they occur in sympatry?

L 285-286: "The Andes rose much later (about 25 MY ago)" is a bit of an oversimplification, with the southern Andes being much older than the northern Andes, and the 'age' of the mountains throughout being strongly dependent on the elevation above sea level that you are discussing. Furthermore, the statement "and were populated by butterflies originating in the eastern parts of the continent'" is also a large over-generalization.

L 287: Andean butterflies? Butterflies in general? Neither statement is generally correct though, since peak diversity in the eastern tropical Andes is usually around 700 m, and in the western Andes a bit higher, and obviously it differs greatly depending on the group, I imagine Riodinidae peak at around 400-500 m for example, in the eastern Andes. More informative here would be a statement about where Lycaenidae diversity peaks.

L 292: 'intrincated' should be 'intricate', plus inclusion of some references to support this statement would be good, e.g. papers on Adelpha, Heliconius, which include broad maps of Neotropical butterfly diversity.

L 293: 'endemic to the valley system' – not really clear what you mean by this. Few species are endemic to individual valleys.

L 295-296: again, some references would be useful, e.g. papers on pronophilines.

L 296-303: more references needed, including more recent references.

L 304-305, reword to clarify and avoid repetition.

L 314-317: but these phenomena explain paraphyletic and polyphyletic species, whereas some monophyletic species with low separation may have been lumped as an artifact of the broad scale of the analysis. It would be nice to quantify how many of the problem BIN sharing was due to these two different phenomena.

L 318-319: very important to make sure you are comparing similar scenarios. For example, a study of barcodes in a single site is more likely to find barcode gaps than one based on the same taxa across a broader geographic area. A study of barcodes in one less diverse site is more likely to find gaps than one in a more diverse site (where sister species are more likely to co-occur). Finally, a study of barcodes in genus may also be more likely to find barcode gaps if sampling within each species is geographically very limited.

L 322-325: this is an interesting hypothesis that could be better supported if it was known how many cases of BIN sharing resulted from hybridization vs low divergence between clades (i.e., biological vs methodological issues, as mentioned above?.

---

## Round 0.2 · Minor Revisions

Two reviewers have now revised your manuscript. Although they are overall happy with the new changes incorporated to the new version, they still raised some minor concerns that would need to be addressed. Please take them into consideration while preparing the revised version of your manuscript.

Reviewer 1 ·

Basic reporting

The manuscript is overall clean and clear. However there are spelling errors and in the manuscript the references are out of chronological order. All of them are marked in yellow in the attached file.
More importantly, in lines 115-116 they say 'A specific objective of this paper is to test the hypothesis that the ability of DNA barcode BINs to discriminate morphologically delimited species decreases in high elevation lineages', but this it is an objective, not an hypothesis. The hypothesis is a proposed explanation made on the basis of some evidence, as starting point for further investigation, so I recommend that the authors make the hypothesis and explain why the DNA barcode decreases its ability to discriminate in high elevation linages.

The results are relevant and deserve to be published after minor corrections and clarifications.

Experimental design

The data and findings of this paper are valuable. This group of butterflies are one of the most complicated for several reasons, and the authors explore the effectiveness of this barcode tool as a first approach to the problem.

They added some statistical test; however, perhaps they should prefer other tests to find out whether or not the variation is significant between their comparisions. I suggest ANOVA or Kruskal Wallis test

Validity of the findings

This contribution is important. The massive data that the authors generated will be important for future research. This is a first approximation to the usefulness of the barcode as a tool to identify species that are first morphologically discriminated. The conclusions are appropriate. Also, this can still improve on the statistical part as suggested above.

Additional comments

It is an important contribution for the taxonomic group and for the high diversity area.

Even so, there are still errors that need to be corrected and the statistical part can still be improved with the suggested tests. The same with the hypothesis in lines 115 and 116, since they are not an explanation of the facts that you observed. All of them are marked in yellow in the attached file.

Annotated reviews are not available for download in order to protect the identity of reviewers who chose to remain anonymous.

·

Basic reporting

The authors have addressed most of my comments from the previous submission, I just added a few new comments or requests for clarification in the manuscript file.

Experimental design

More detail was added about the RESL analysis, but still it is difficult to understand what a priori settings are needed for this analysis, and since the paper refers to these settings in a couple of places, I think a more detailed description would be useful of how the analysis works and what a priori settings decisions need to be made.

Validity of the findings

The authors have added some additional references and discussion about Andean butterfly biogeography, and I still think that the conclusions are largely sound, if a bit qualitative, but it is the authors' decision whether or not to delve further into their findings in this paper, or leave it for a future study. I am happy with either one.

Additional comments

Overall, this is a really remarkable study which will provides a huge dataset of great value to Neotropical butterfly researchers, and I think the revised version addresses most of my concerns/queries, except for notes in the attached manuscript.

---

## Round 0.3 · accepted · Accept

Dear Dr. Prieto,

Two reviewers have now assessed the revised version of your manuscript. They are both happy with the changes and responses provided and I am happy to inform you that your manuscript is now ready for publication in PeerJ. Congratulations!

·

Basic reporting

No comment.

Experimental design

No comment.

Validity of the findings

No comment.

Additional comments

The authors have addressed my comments from the previous review and I feel the manuscript should be accepted as it is.